# Challenges in Explaining Representational Similarity through Identifiability

**Beatrix M. G. Nielsen**[*,1]       **Luigi Gresele**[†,2]       **Andrea Dittadi**[†,3,4,5,6]

[1] Technical University of Denmark, [2] University of Copenhagen, [3] Helmholtz AI, [4] TU Munich,
[5] Munich Center for Machine Learning (MCML), [6] MPI for Intelligent Systems, Tübingen

## Abstract

The phenomenon of different deep learning models producing similar data representations has garnered significant attention, raising the question of why such representational similarity occurs. Identifiability theory offers a partial explanation: for a broad class of discriminative models, including many popular in representation learning, those assigning equal likelihood to the observations yield representations that are equal up to a linear transformation, if a suitable diversity condition holds. In this work, we identify two key challenges in applying identifiability theory to explain representational similarity. First, the assumption of exact likelihood equality is rarely satisfied by practical models trained with different initializations. To address this, we describe how the representations of two models deviate from being linear transformations of each other, based on their difference in log-likelihoods. Second, we demonstrate that even models with similar and near-optimal loss values can produce highly dissimilar representations due to an underappreciated difference between loss and likelihood. Our findings highlight key open questions and point to future research directions for advancing the theoretical understanding of representational similarity.

## 1   Introduction

There is ample evidence of similarity in the representations learned by different deep learning models (Lenc and Vedaldi, 2015; Kornblith et al., 2019; Bansal et al., 2021; Ding et al., 2021), which has given rise to conjectures on why the phenomenon occurs (Huh et al., 2024; Teney et al., 2024) as well as work on exploiting it (Moschella et al., 2022; Cannistraci et al., 2023; Maiorca et al., 2024), for example for model stitching (Lenc and Vedaldi, 2015; Bansal et al., 2021). One instance of representational similarity, perhaps the simplest and most fundamental one, is when different models trained on the same data and with the same learning objective produce representations that are equal up to a simple transformation, for example linear.[3] For highly nonlinear models, it is not obvious that this should occur: in fact, work on nonlinear independent component analysis (ICA) offers many examples where different models, despite assigning equal likelihood to the data, yield representations related by nonlinear transformations (see, e.g., (Hyvärinen and Pajunen, 1999)). However, Roeder et al. (2021) and Khemakhem et al. (2020) show that for nonlinear models in a broad discriminative class—including several models popular in representation learning, those which assign equal likelihood to the observations extract representations which *are* linear transformations

---

[*]Corresponding author: bmgi@dtu.dk.

[†]Equal advising.

[3]though the representational similarity phenomenon appears to be broader, and it encompasses models trained with *different* learning objectives and datasets, see, e.g., (Moschella et al., 2022; Huh et al., 2024).

Workshop on Unifying Representations in Neural Models (UniReps) at NeurIPS 2024, Extended Abstract.

of one another, provided a suitable *diversity* condition is satisfied. One might therefore be inclined to use these findings to account for some of the empirical observations of representational similarity.

In this work, we identify two key challenges in using existing identifiability theory to explain representational similarity. Firstly, identifiability results require equality of the likelihoods of the considered models as a premise: in practice, this assumption is rarely fulfilled for trained models, since even initializing with different random seeds will typically result in models with likelihoods which might be close, but not equal. It would therefore be desirable to relax the assumption of equality and explore whether the results in (Roeder et al., 2021; Khemakhem et al., 2020) can be extended to prove *approximate* representational similarity[4] for models achieving *close to equal* likelihoods. Furthermore, as we will show in Section 4, minimizing the loss is not the same as minimizing the difference of log-likelihoods, which means that models with close to optimal loss can have representations which are very dissimilar. By pointing out these challenges, we hope to inspire further research on the theoretical underpinnings of representational similarity.

Our contributions can be summarized as follows:

- In Section 3, we prove a relation between the difference of log-likelihoods entailed by two models and their extracted representations. This generalizes the analysis in (Roeder et al., 2021; Khemakhem et al., 2020), which requires vanishing log-likelihood difference.

- In Section 4, we introduce a construction showing that it is not sufficient that both considered models have close to optimal (zero) loss for their representations to become similar.

## 2 Preliminaries

**Model Class.**   We will consider a model class defined by the probability of a label, $\mathbf{y}$, given an input, $\mathbf{x}$, from a domain, $\mathcal{X}$, and a collection of possible targets, $\mathcal{S}$, where we must have $\mathbf{y} \in \mathcal{S}$.

$$p_{\boldsymbol{\theta}}(\mathbf{y}|\mathbf{x}, \mathcal{S}) = \frac{\exp(\mathbf{f}_{\boldsymbol{\theta}}(\mathbf{x})^{\top} \mathbf{g}_{\boldsymbol{\theta}}(\mathbf{y}))}{\sum_{\mathbf{y}' \in \mathcal{S}} \exp(\mathbf{f}_{\boldsymbol{\theta}}(\mathbf{x})^{\top} \mathbf{g}_{\boldsymbol{\theta}}(\mathbf{y}'))} \tag{1}$$

where $\boldsymbol{\theta}$ is the parameters of the model. We will often write "the model $\boldsymbol{\theta}$" as short for the model with parameters $\boldsymbol{\theta}$. For two models, $\boldsymbol{\theta}^*$ and $\boldsymbol{\theta}'$, we write $\mathbf{f}^*$ and $\mathbf{f}'$ for their embedding functions and $\mathbf{g}^*$ and $\mathbf{g}'$ for their unembedding functions, following the terminology in (Park et al., 2023). In the following, we let the codomain of $\mathbf{f}$ and $\mathbf{g}$ be $\mathbb{R}^M$. So the embedding functions $\mathbf{f} : \mathcal{X} \to \mathbb{R}^M$ take inputs into $\mathbb{R}^M$, and the unembedding functions $\mathbf{g} : \mathcal{S} \to \mathbb{R}^M$ take the labels into $\mathbb{R}^M$. This model class is the same as the one considered in (Roeder et al., 2021): it is particularly interesting because many common models and pre-training objectives can be written in this form, for example autoregressive language models and a common supervised classification objective. For more examples, see (Roeder et al., 2021, Section 4 & Appendix D).

**Diversity Condition.**   We define the diversity condition like in (Khemakhem et al., 2020).

**Definition 1** (Diversity condition). We say that a model $\boldsymbol{\theta}'$ from the model class Eq. (1) satisfies the diversity condition for $\mathbf{g}'$ if there exists $\mathbf{y}_0, ..., \mathbf{y}_M \in \mathcal{S}$ such that the $M$ vectors $\{\mathbf{g}'(\mathbf{y}_i) - \mathbf{g}'(\mathbf{y}_0)\}_{i=1}^{M}$ are linearly independent. Similarly, we say that a model satisfies the diversity condition for $\mathbf{f}'$ if there exists $\mathbf{x}_0, ..., \mathbf{x}_M \in \mathcal{X}$ such that the $M$ vectors $\{\mathbf{f}'(\mathbf{x}_i) - \mathbf{f}'(\mathbf{x}_0)\}_{i=1}^{M}$ are linearly independent.

This assumption, or variations thereof, plays a crucial role in the identifiability results in, e.g., (Roeder et al., 2021), (Khemakhem et al., 2020) and (Lachapelle et al., 2023).

**Identifiability Result.**   For our purposes, the identifiability results found in (Khemakhem et al., 2020), (Roeder et al., 2021), and (Lachapelle et al., 2023) can be summarized in the following statement (a detailed proof can be found in Appendix B).

**Theorem 1.** *Let $\boldsymbol{\theta}^*$ be a model of the form in Eq. (1) and satisfying the diversity condition on $\mathbf{f}^*$ and $\mathbf{g}^*$. Assume $\boldsymbol{\theta}'$ is another model of the same form. Then*

$$p_{\boldsymbol{\theta}^*} = p_{\boldsymbol{\theta}'} \implies \boldsymbol{\theta}^* \sim_L \boldsymbol{\theta}', \tag{2}$$

---

[4]Buchholz and Schölkopf (2024) also consider approximate identifiability, but for a different model class.

*where following ([Lachapelle et al., 2023](#)), the equivalence relation is defined by*

$$\boldsymbol{\theta}^* \sim_L \boldsymbol{\theta}' \iff \begin{cases} \mathbf{f}^*(\mathbf{x}) = \mathbf{A}\mathbf{f}'(\mathbf{x}) \\ \mathbf{g}^*(\mathbf{y}) = \mathbf{A}^{-\top}\mathbf{g}'(\mathbf{y}) + \mathbf{b} \end{cases}, \tag{3}$$

*where $\mathbf{A}$ is an invertible matrix and $\mathbf{b}$ is a vector.*

**Measuring Similarity with Canonical Correlation Analysis.** It is possible to define a measure of how close two sets of vectors are to being linear transformations of each other, based on Canonical Correlation Analysis (CCA) ([Hotelling, 1936](#)). Given two random vectors $\mathbf{z} \in \mathbb{R}^N$ and $\mathbf{w} \in \mathbb{R}^M$, CCA finds vectors $\mathbf{s}_k$ and $\mathbf{t}_k$, where $k \leq \min(N, M)$, such that the Pearson correlation $\rho_k = \text{corr}\left(\mathbf{s}_k^\top \mathbf{z}, \mathbf{t}_k^\top \mathbf{w}\right)$ is maximized, with the constraint that $\mathbf{s}_k^\top \mathbf{z}, \mathbf{s}_j^\top \mathbf{z}$ and $\mathbf{t}_k^\top \mathbf{w}, \mathbf{t}_j^\top \mathbf{w}$ are linearly independent for $k \neq j$. In our setting, we would like to measure how close the embeddings and unembeddings from two models $\boldsymbol{\theta}^*, \boldsymbol{\theta}'$ are to being linear transformations of each other. Our vectors will thus be $\mathbf{f}^*(\mathbf{x}), \mathbf{f}'(\mathbf{x})$ or $\mathbf{g}^*(\mathbf{y}), \mathbf{g}'(\mathbf{y})$ for inputs $\mathbf{x}$ and labels $\mathbf{y}$ (thereby $N = M$). As our similarity score, we will use the mean canonical correlation, as in ([Klabunde et al., 2023](#)):

$$m_{\text{CCA}}(\mathbf{z}, \mathbf{w}) = \frac{1}{M} \sum_k \rho_k$$

If $m_{\text{CCA}}(\mathbf{z}, \mathbf{w}) = 1$, $\mathbf{z}$ and $\mathbf{w}$ are linear transformations of each other (see Appendix A).

**The Difference Between Loss and Likelihood.** The distinction between *loss* and *likelihood* is crucial, since we would like to understand whether different models achieving small *loss* (i.e., close to optimal) will extract similar representations. We say that two models, $\boldsymbol{\theta}^*, \boldsymbol{\theta}'$, have equal *likelihood* if $p_{\boldsymbol{\theta}^*}(\mathbf{y}|\mathbf{x}, \mathcal{S}) = p_{\boldsymbol{\theta}'}(\mathbf{y}|\mathbf{x}, \mathcal{S})$ for all $\mathbf{x} \in \mathcal{X}$ and $\mathbf{y} \in \mathcal{S}$. When training such models, the *loss* we optimize is $\mathbb{E}_{(\mathbf{x},\mathbf{y}) \sim q_{\mathcal{D}}}[-\log(p_{\boldsymbol{\theta}}(\mathbf{y}|\mathbf{x}, \mathcal{S}))]$, where $(\mathbf{x}, \mathbf{y})$ is an input–label pair from the data distribution $q_{\mathcal{D}}$. Note that the identifiability results by, e.g., [Roeder et al. (2021)](#) and [Khemakhem et al. (2020)](#) require that the *likelihood* entailed by the two models, $\boldsymbol{\theta}^*$ and $\boldsymbol{\theta}'$, should be equal. However, the observed data distribution $q_{\mathcal{D}}(\mathbf{y}|\mathbf{x})$ will likely not include all possible combinations of inputs values $\mathbf{x} \in \mathcal{X}$ and label values in $\mathcal{S}$. For example, in the case of an autoregressive language model, given a sequence of words $\mathbf{x}$, many among the potential next token-candidates $\mathbf{y}$ would result in nonsensical sentences, thereby having a low probability of appearing in the training data. Because of this, for each $\mathbf{x}$, minimizing the loss does not uniquely constrain the conditional likelihood for all targets in $\mathcal{S}$. In short, equal loss does not mean equal likelihood. One might nevertheless think that if the loss is close enough to optimal, and there is very little density assigned to improbable targets, we would still get models with representations which are *close* to being linear transformations of each other. However, as we show in Section 4, this is not the case.

## 3 Close-to-Identifiability Result

As a first step towards generalizing the results in ([Roeder et al., 2021](#); [Khemakhem et al., 2020](#)), we prove that for models as in Equation (1), it holds that the representations extracted by a model $\boldsymbol{\theta}^*$ can be written as a linear transformation of those from another model, $\boldsymbol{\theta}'$, plus an error term which can be seen as the non-linear part of the relationship between the functions. When the likelihoods of the two models are equal, the error term vanishes, and we recover the results of Theorem 1.

**Theorem 2.** *Let $\boldsymbol{\theta}^*$ and $\boldsymbol{\theta}'$ be two models, and let $\boldsymbol{\theta}^*$ satisfy the diversity condition (Definition 1) for both $\mathbf{f}^*$ and $\mathbf{g}^*$. Let $\mathbf{y}_0, ..., \mathbf{y}_M$ be the $\mathbf{y}_i$s from the diversity condition on $\mathbf{g}^*$. Let $\mathbf{L}^*$ be the matrix with columns $\mathbf{g}^*(\mathbf{y}_i) - \mathbf{g}^*(\mathbf{y}_0)$ and $\mathbf{L}'$ the matrix with columns $\mathbf{g}'(\mathbf{y}_i) - \mathbf{g}'(\mathbf{y}_0)$. Then*

$$\mathbf{f}^*(\mathbf{x}) = \mathbf{A}\mathbf{f}'(\mathbf{x}) + \mathbf{h}_{\mathbf{f}^*}(\mathbf{x}) \tag{4}$$

$$\mathbf{g}^*(\mathbf{y}) = \mathbf{B}\mathbf{g}'(\mathbf{y}) + \mathbf{h}_{\mathbf{g}^*}(\mathbf{y}) \tag{5}$$

*where $\mathbf{A} = \mathbf{L}^{*-\top}\mathbf{L}'^{\top}$, $\mathbf{h}_{\mathbf{f}^*}(\mathbf{x}) = \mathbf{L}^{*-\top}\boldsymbol{\epsilon}_{\mathbf{y}}(\mathbf{x})$, and $\boldsymbol{\epsilon}_{\mathbf{y}}(\mathbf{x})$ is a vector function with each entry equal to $\epsilon_{\mathbf{y}i}(\mathbf{x}) = \mathbf{f}^*(\mathbf{x})^\top \mathbf{g}^*(\mathbf{y}_i) - \mathbf{f}'(\mathbf{x})^\top \mathbf{g}'(\mathbf{y}_i) + \mathbf{f}'(\mathbf{x})^\top \mathbf{g}'(\mathbf{y}_0) - \mathbf{f}^*(\mathbf{x})^\top \mathbf{g}^*(\mathbf{y}_0)$. So $\boldsymbol{\epsilon}_{\mathbf{y}}(\mathbf{x})$ is a function of $\mathbf{x}$ using the $\mathbf{y}_i$s from the diversity condition on $\mathbf{g}^*$. $\mathbf{B}$ will be a similar product of matrices, only using the diversity condition on $\mathbf{f}^*$. Also, $\mathbf{h}_{\mathbf{g}^*}(\mathbf{y})$ will contain a $\boldsymbol{\epsilon}_{\mathbf{x}}(\mathbf{y})$ which is a function of $\mathbf{y}$ using the $\mathbf{x}_i$s from the diversity condition on $\mathbf{f}^*$.*

The proof can be found in Appendix C. A consequence of Theorem 2 is that there is the following relationship between the embeddings $\mathbf{f}^*(\mathbf{x}), \mathbf{f}'(\mathbf{x})$:

$$\mathbf{f}^*(\mathbf{x}) = \mathbf{L}^{*-\top} \left( \mathbf{L}'^{\top} \mathbf{f}'(\mathbf{x}) + \boldsymbol{\epsilon}_y(\mathbf{x}) \right) \tag{6}$$

and a similar one for the unembeddings $\mathbf{g}^*(\mathbf{y}), \mathbf{g}'(\mathbf{y})$. This equation shows that whether we can say that $\mathbf{f}'(\mathbf{x})$ is close to being a linear transformation of $\mathbf{f}^*(\mathbf{x})$ depends on the relative size of $\boldsymbol{\epsilon}_\mathbf{y}(\mathbf{x})$ compared to $\mathbf{L}'^{\top} \mathbf{f}'(\mathbf{x})$. If $\|\boldsymbol{\epsilon}_\mathbf{y}(\mathbf{x})\| << \|\mathbf{L}'^{\top} \mathbf{f}'(\mathbf{x})\|$, then the representations will be close to linear transformations of each other. Since $\boldsymbol{\epsilon}_y(\mathbf{x})$ and $\boldsymbol{\epsilon}_x(\mathbf{y})$ can be expressed in terms of differences of log-likelihoods entailed by the two models (see Appendix C), we see that they will be small if the models assign likelihoods which are close to equal to the observations. In particular, we see that for the embedding representations to be close, we need log-likelihoods to be close for all $\mathbf{x}$ and for all the $\mathbf{y}_i$'s from the diversity condition on $\mathbf{g}$. Conversely, for the $\mathbf{g}$ representations to be close, we need log-likelihoods to be close for all $\mathbf{y}$ and for all the $\mathbf{x}_i$'s from the diversity condition on $\mathbf{f}$. If the distributions for the models are equal, $\boldsymbol{\epsilon}_y(\mathbf{x})$ and $\boldsymbol{\epsilon}_x(\mathbf{y})$ will be zero, and we recover the identifiability result of Theorem 1. However, optimizing $p^*(\mathbf{y}|\mathbf{x}, \mathcal{S})$ and $p'(\mathbf{y}|\mathbf{x}, \mathcal{S})$ for the correct label, $\mathbf{y}$, is not enough to make this difference of log-likelihoods small. Below we present an example of this.

## 4   Example of Close to Zero Loss where Representations are Dissimilar

In this example, we have $M = 2$ and a classification task with four labels, $\mathbf{y}_0, \mathbf{y}_1, \mathbf{y}_2, \mathbf{y}_3$. The example relies on the fact that if we fix non-zero angles between the unembedding vectors, and we let the embedding representations be closer to the correct label in terms of angle than the incorrect ones, then we can make the likelihood of the correct label arbitrarily close to 1 by only changing the lengths of the unembedding vectors (see Appendix E, and Appendix D for more details).

For the first model $\boldsymbol{\theta}'$, we let the the angle between the unembeddings be very small, and the lengths of the unembeddings be very large. We generate our embedding vectors such that they have Euclidean norms larger than 1 and such that they are very close in terms of angle to the unembeddings with the correct label (see Fig. 1). For the second model, $\boldsymbol{\theta}^*$, we spread out the unembedding representations such that three are on the axes and one is slightly off. For the embedding representations, we place them such that they are closer in terms of angle to the unembedding with the correct label, but more spread out than the ones from model $\boldsymbol{\theta}'$ (see Fig. 1).

We can now calculate the negative log-likelihood ($NLL$) for these two models using Eq. (1). For model $\boldsymbol{\theta}'$, we get $NLL' \approx 9 \cdot 10^{-10}$ and for model $\boldsymbol{\theta}^*$, we get $NLL^* \approx 7 \cdot 10^{-10}$. In fact, for the correct labels, we get a small difference in log-likelihood for all datapoints. The maximal difference for the two models is $8 \cdot 10^{-7}$. As mentioned above, we could make this loss arbitrarily small, by increasing the lengths of the unembeddings. Now both of these models have close to zero loss, and their loss thus is close to equal:

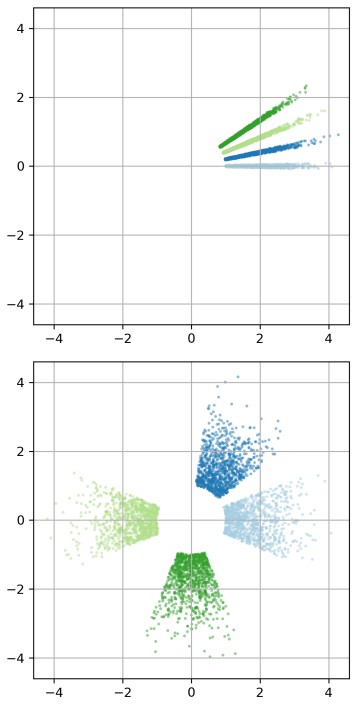

Figure 1: Embeddings $\mathbf{f}'(\mathbf{x})$ (above) and $\mathbf{f}^*(\mathbf{x})$ (below). Unembeddings are not shown in the figure.

however, they are very far from being linear transformations of each other. For example, the mean canonical correlation between the embedding representations is $m_{\mathrm{CCA}}(\mathbf{f}'(\mathbf{x}), \mathbf{f}^*(\mathbf{x})) \approx 0.42$, which is very far from the value of 1 which would indicate a perfect linear relationship. See Appendix D for further insights on this value of $m_{\mathrm{CCA}}$ and the degree of dissimilarity it indicates. It is possible to construct an example with even smaller $m_{\mathrm{CCA}}$, for example by making the angles smaller and the lengths longer for $\mathbf{g}'(\mathbf{y})$ for model $\boldsymbol{\theta}'$, while keeping the other model $\boldsymbol{\theta}^*$ as it is.

# 5 Conclusion

We showed that the representations (embeddings and unembeddings) extracted by two models of the form in Eq. (1) will be close to being linear transformations of each other for all $\mathbf{x}$ and $\mathbf{y}$, if the log-likelihoods entailed by the two models are close for all $\mathbf{x}$ and $\mathbf{y}$. We also introduced a construction to show that, for the representations of two models to be close to equivalent, it is not sufficient that the losses of both models are close to each other and small. These results point to interesting questions for future research: for example, how a non-vanishing difference in log-likelihood can be connected to a measurement of representational similarity; and under what additional assumptions similarity should be expected if the difference in *expected log-likelihood* or in *loss* is non-vanishing.

**Acknowledgments**

The authors would like to thank Emanuele Marconato, Julius von Kügelgen and Adrián Javaloy for valuable discussions. This work was supported by the Danish Pioneer Centre for AI, DNRF grant number P1. L.G. was supported by the Danish Data Science Academy (DDSA), Grant ID: 2023-1250.

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

# Part I

# Appendix

## Table of Contents

## A   More About CCA

We show that if $\mathbf{f}^*(\mathbf{x}) = \mathbf{A}\mathbf{f}'(\mathbf{x})$, where $\mathbf{A}$ is an invertible matrix, then $m_{\text{CCA}}(\mathbf{f}^*(\mathbf{x}), \mathbf{f}'(\mathbf{x})) = 1$ and that if $m_{\text{CCA}}(\mathbf{f}^*(\mathbf{x}), \mathbf{f}'(\mathbf{x})) = 1$, then $\mathbf{f}^*(\mathbf{x}) = \mathbf{A}\mathbf{f}'(\mathbf{x}) + \mathbf{b}$ where $\mathbf{A}$ is an invertible matrix and $\mathbf{b}$ is a vector. So if we assume that the representations are centered, we have that if $m_{\text{CCA}}(\mathbf{f}^*(\mathbf{x}), \mathbf{f}'(\mathbf{x})) = 1$, then $\mathbf{f}^*(\mathbf{x}) = \mathbf{A}\mathbf{f}'(\mathbf{x})$.

*Proof.* Assume $\mathbf{f}^*(\mathbf{x}) = \mathbf{A}\mathbf{f}'(\mathbf{x})$, where $\mathbf{A}$ is an invertible matrix. Then the $k$'th entry of $\mathbf{f}^*(\mathbf{x})$ is equal to

$$\mathbf{f}^*(\mathbf{x})_k = \mathbf{A}_k \mathbf{f}'(\mathbf{x})$$

where $\mathbf{A}_k$ is the $k$'th row of $\mathbf{A}$. Let

$$\mathbf{s}_k = \mathbf{e}_k \qquad , \qquad \mathbf{t}_k = \mathbf{A}_k^\top \tag{7}$$

where $\mathbf{e}_k$ is the standard basis vector, we get the vectors used by CCA. Note that since $\mathbf{A}$ is invertible, the rows $\mathbf{A}_k$ are linearly independent. Then

$$\rho_k = \text{corr}\left(\mathbf{s}_k^\top \mathbf{f}^*(\mathbf{x}), \mathbf{t}_k^\top \mathbf{f}'(\mathbf{x})\right) = 1 \tag{8}$$

for all $k \in \{1, ..., M\}$. Thus $m_{\text{CCA}}(\mathbf{f}^*(\mathbf{x}), \mathbf{f}'(\mathbf{x})) = 1$.

Assume now that $m_{\text{CCA}}(\mathbf{f}^*(\mathbf{x}), \mathbf{f}'(\mathbf{x})) = 1$. Then there exist from CCA, vectors $\mathbf{s}_k, \mathbf{t}_k$ such that

$$\rho_k = \text{corr}\left(\mathbf{s}_k^\top \mathbf{f}^*(\mathbf{x}), \mathbf{t}_k^\top \mathbf{f}'(\mathbf{x})\right) = 1 \tag{9}$$

for all $k \in \{1, ..., M\}$. Therefore,

$$\mathbf{s}_k^\top \mathbf{f}^*(\mathbf{x}) = \frac{\sigma(\mathbf{s}_k^\top \mathbf{f}^*(\mathbf{x}))}{\sigma(\mathbf{t}_k^\top \mathbf{f}'(\mathbf{x}))} \mathbf{t}_k^\top \mathbf{f}'(\mathbf{x}) + c \tag{10}$$

where $\sigma$ is the standard deviation and $c$ is some constant. Therefore, if we stack these equations, we get

$$\mathbf{S}\mathbf{f}^*(\mathbf{x}) = \mathbf{T}_\sigma \mathbf{f}'(\mathbf{x}) + \mathbf{c} \tag{11}$$

where $\mathbf{S}$ and $\mathbf{T}_\sigma$ are invertible matrices, therefore we get

$$\mathbf{f}^*(\mathbf{x}) = \mathbf{S}^{-1}\mathbf{T}_\sigma \mathbf{f}'(\mathbf{x}) + \mathbf{S}^{-1}\mathbf{c} \tag{12}$$

so letting $\mathbf{A} = \mathbf{S}^{-1}\mathbf{T}_\sigma$ and $\mathbf{b} = \mathbf{S}^{-1}\mathbf{c}$, we have the result.

$\square$

## B   Identifiability Result and Proof

We here present a merging of the identifiability results found in (Khemakhem et al., 2020), (Roeder et al., 2021) and (Lachapelle et al., 2023). Let $\boldsymbol{\theta}^*$ be a model satisfying the diversity condition on $\mathbf{f}^*$ and $\mathbf{g}^*$ (like in (Khemakhem et al., 2020) and not (Roeder et al., 2021)). Assume $\boldsymbol{\theta}'$ is another model. Then

$$p_{\boldsymbol{\theta}^*} = p_{\boldsymbol{\theta}'} \implies \boldsymbol{\theta}^* \sim_L \boldsymbol{\theta}' \tag{13}$$

where the equivalence relation is defined as in (Lachapelle et al., 2023) by

$$\boldsymbol{\theta}^* \sim_L \boldsymbol{\theta}' \iff \begin{cases} \mathbf{f}^*(\mathbf{x}) = \mathbf{A}\mathbf{f}'(\mathbf{x}) \\ \mathbf{g}^*(\mathbf{y}) = \mathbf{A}^{-\top}\mathbf{g}'(\mathbf{y}) + \mathbf{b} \end{cases} \tag{14}$$

where $\mathbf{A}$ is an invertible matrix and $\mathbf{b}$ is a vector.

*Proof.* We first prove that $p^* = p' \implies \mathbf{f}^*(\mathbf{x}) = \mathbf{A}\mathbf{f}'(\mathbf{x})$ for $\mathbf{A}$ invertible.

Assume the models have equal likelihoods. Let the codomains of $\mathbf{f}^*, \mathbf{g}^*, \mathbf{f}', \mathbf{g}'$ be in $\mathbb{R}^M$. Let $Z^*(\mathbf{x}, \mathcal{S}) = \sum_{\mathbf{y}_j \in \mathcal{S}} \exp(\mathbf{f}^*(\mathbf{x})^\top \mathbf{g}^*(\mathbf{y}_j))$, and similarly for $Z'(\mathbf{x}, \mathcal{S})$. Then

$$p^*(\mathbf{y}|\mathbf{x}, \mathcal{S}) = p'(\mathbf{y}|\mathbf{x}, \mathcal{S}) \tag{15}$$

$$\mathbf{f}^*(\mathbf{x})^\top \mathbf{g}^*(\mathbf{y}) - \log(Z^*(\mathbf{x}, \mathcal{S})) = \mathbf{f}'(\mathbf{x})^\top \mathbf{g}'(\mathbf{y}) - \log(Z'(\mathbf{x}, \mathcal{S})) \tag{16}$$

for all $\mathbf{y}$. In particular, it is true for the $M + 1$ $\mathbf{y}$s, $y_0, ..., y_M$ which exist according to the diversity condition on $\mathbf{g}^*$. So we can write up $M + 1$ equations of this kind. If we subtract the equation with $y_0$ from these, we are left with $M$ equations of the form

$$\mathbf{f}^*(\mathbf{x})^\top \mathbf{g}^*(\mathbf{y}_i) - \mathbf{f}^*(\mathbf{x})^\top \mathbf{g}^*(\mathbf{y}_0) + \log(Z^*(\mathbf{x}, \mathcal{S})) - \log(Z^*(\mathbf{x}, \mathcal{S})) \tag{17}$$

$$= \mathbf{f}'(\mathbf{x})^\top \mathbf{g}'(\mathbf{y}_i) - \mathbf{f}'(\mathbf{x})^\top \mathbf{g}'(\mathbf{y}_0) + \log(Z'(\mathbf{x}, \mathcal{S})) - \log(Z'(\mathbf{x}, \mathcal{S})) \tag{18}$$

$$\mathbf{f}^*(\mathbf{x})^\top \mathbf{g}^*(\mathbf{y}_i) - \mathbf{f}^*(\mathbf{x})^\top \mathbf{g}^*(\mathbf{y}_0) = \mathbf{f}'(\mathbf{x})^\top \mathbf{g}'(\mathbf{y}_i) - \mathbf{f}'(\mathbf{x})^\top \mathbf{g}'(\mathbf{y}_0) \tag{19}$$

$$\mathbf{f}^*(\mathbf{x})^\top (\mathbf{g}^*(\mathbf{y}_i) - \mathbf{g}^*(\mathbf{y}_0)) = \mathbf{f}'(\mathbf{x})^\top (\mathbf{g}'(\mathbf{y}_i) - \mathbf{g}'(\mathbf{y}_0)) \tag{20}$$

$$(\mathbf{g}^*(\mathbf{y}_i) - \mathbf{g}^*(\mathbf{y}_0))^\top \mathbf{f}^*(\mathbf{x}) = (\mathbf{g}'(\mathbf{y}_i) - \mathbf{g}'(\mathbf{y}_0))^\top \mathbf{f}'(\mathbf{x}) \tag{21}$$

$$\tag{22}$$

Let $\mathbf{L}^*$ be the matrix which has $\mathbf{g}^*(\mathbf{y}_i) - \mathbf{g}^*(y_0)$ as columns and $\mathbf{L}'$ be the matrix which has $\mathbf{g}'(\mathbf{y}_i) - \mathbf{g}'(y_0)$ as columns. We can then stack the equations to get

$$\mathbf{L}^{*T}\mathbf{f}^*(\mathbf{x}) = \mathbf{L}'^\top \mathbf{f}'(\mathbf{x}) \tag{23}$$

and since $\mathbf{L}^*$ is invertible,

$$\mathbf{f}^*(\mathbf{x}) = \mathbf{L}^{*-\top}\mathbf{L}'^\top \mathbf{f}'(\mathbf{x}) \tag{24}$$

If we set $\mathbf{A} = (\mathbf{L}'\mathbf{L}^{*-1})^\top$, we only need to show that $\mathbf{A}$ is invertible. Using the diversity condition on $\mathbf{f}^*$, we pick points $\mathbf{x}_0, ..., x_M$ such that $\mathbf{f}^*(\mathbf{x}_i) - \mathbf{f}^*(\mathbf{x}_0)$ are linearly independent. Let $\mathbf{N}^*$ be the matrix with $\mathbf{f}^*(\mathbf{x}_i) - \mathbf{f}^*(\mathbf{x}_0)$ as columns and $\mathbf{N}'$ be the matrix with $\mathbf{f}'(\mathbf{x}_i) - \mathbf{f}'(\mathbf{x}_0)$ as columns. Then

$$\mathbf{N}^* = \mathbf{A}\mathbf{N}' \tag{25}$$

Since we know that for any two matrices, $\mathbf{B}, \mathbf{C}$, $\text{rank}(\mathbf{BC}) \leq \min(\text{rank}(\mathbf{B}), \text{rank}(\mathbf{C}))$, and $\mathbf{N}^*$ has rank $M$, we see that $\mathbf{A}$ and $\mathbf{N}'$ must both also have rank $M$. Thus, $\mathbf{A}$ is invertible.

Next we prove that $p^* = p' \implies \mathbf{g}^*(\mathbf{y}) = \mathbf{A}^{-\top}\mathbf{g}'(\mathbf{y}) + \mathbf{b}$ for $\mathbf{A}$ invertible and $\mathbf{b}$ a vector.

As before we have that

$$\mathbf{f}^*(\mathbf{x})^\top \mathbf{g}^*(\mathbf{y}) - \log(Z^*(\mathbf{x}, \mathcal{S})) = \mathbf{f}'(\mathbf{x})^\top \mathbf{g}'(\mathbf{y}) - \log(Z'(\mathbf{x}, \mathcal{S})) \tag{26}$$

holds for all $\mathbf{x}$. In particular, it is true for the $M + 1$ $\mathbf{x}$'s, $\mathbf{x}_0, ..., x_M$ which exist according to the diversity condition on $\mathbf{f}^*$. So we can write up $M + 1$ equations of this kind. If we subtract the equation with $\mathbf{x}_0$ from these, we are left with $M$ equations of the form

$$\mathbf{f}^*(\mathbf{x}_i)^\top \mathbf{g}^*(\mathbf{y}) - \mathbf{f}^*(\mathbf{x}_0)^\top \mathbf{g}^*(\mathbf{y}) + \log(Z^*(\mathbf{x}_0, \mathcal{S})) - \log(Z^*(\mathbf{x}_i, \mathcal{S})) \tag{27}$$

$$= \mathbf{f}'(\mathbf{x}_i)^\top \mathbf{g}'(\mathbf{y}) - \mathbf{f}'(\mathbf{x}_0)^\top \mathbf{g}'(\mathbf{y}) + \log(Z'(\mathbf{x}_0, \mathcal{S})) - \log(Z'(\mathbf{x}_i, \mathcal{S})) \tag{28}$$

$$(\mathbf{f}^*(\mathbf{x}_i) - \mathbf{f}^*(\mathbf{x}_0))^\top \mathbf{g}^*(\mathbf{y}) = (\mathbf{f}'(\mathbf{x}_i) - \mathbf{f}'(\mathbf{x}_0))^\top \mathbf{g}'(\mathbf{y}) + c_i \tag{29}$$

where

$$c_i = \log\left(\frac{Z'(\mathbf{x}_0, \mathcal{S})}{Z^*(\mathbf{x}_0, \mathcal{S})}\right) + \log\left(\frac{Z^*(\mathbf{x}_i, \mathcal{S})}{Z'(\mathbf{x}_i, \mathcal{S})}\right) \tag{30}$$

Let $\mathbf{N}^*$ be the matrix with $\mathbf{f}^*(\mathbf{x}_i) - \mathbf{f}^*(\mathbf{x}_0)$ as columns, let $\mathbf{N}'$ be the matrix with $\mathbf{f}'(\mathbf{x}_i) - \mathbf{f}'(\mathbf{x}_0)$ as columns and let $\mathbf{c}$ be the vector with $c_i$ as entries. Then since $\mathbf{N}^*$ is invertible

$$\mathbf{N}^{*T}\mathbf{g}^*(\mathbf{y}) = \mathbf{N}'^\top \mathbf{g}'(\mathbf{y}) + \mathbf{c} \tag{31}$$

$$\mathbf{g}^*(\mathbf{y}) = \mathbf{N}^{*-\top}\mathbf{N}'^\top \mathbf{g}'(\mathbf{y}) + \mathbf{N}^{*-\top}\mathbf{c} \tag{32}$$

Since we found before that $A$ is invertible and

$$\mathbf{A}^{-1}\mathbf{N}^* = \mathbf{N}' \tag{33}$$

we have that

$$\mathbf{g}^*(\mathbf{y}) = \mathbf{N}^{*-\top}\mathbf{N}'^\top \mathbf{g}'(\mathbf{y}) + \mathbf{N}^{*-\top}\mathbf{c} \tag{34}$$

$$\mathbf{g}^*(\mathbf{y}) = \mathbf{N}^{*-\top}(\mathbf{A}^{-1}\mathbf{N}^*)^\top \mathbf{g}'(\mathbf{y}) + \mathbf{N}^{*-\top}\mathbf{c} \tag{35}$$

$$\mathbf{g}^*(\mathbf{y}) = \mathbf{N}^{*-\top}\mathbf{N}^{*T}A^{-\top}\mathbf{g}'(\mathbf{y}) + \mathbf{N}^{*-\top}\mathbf{c} \tag{36}$$

$$\mathbf{g}^*(\mathbf{y}) = \mathbf{A}^{-\top}\mathbf{g}'(\mathbf{y}) + \mathbf{b} \tag{37}$$

where $\mathbf{b} = \mathbf{N}^{*-\top}\mathbf{c}$, and we have the result. $\qquad\square$

## C  Close-to-Identifiability Result and Full Proof

We here provide the statement and proof of our main contribution. We show that for models as in Equation (1), the representations extracted by a model $\boldsymbol{\theta}^*$ can be written as a linear transformation of those from another model, $\boldsymbol{\theta}'$, plus an error term which can be seen as the non-linear part of the relationship between the functions. When the likelihoods of the two models are equal, the error term vanishes, and we recover the results of Theorem 1. Moreover, the connection we present between representations and difference in log-likelihoods can also be seen as a first step towards a "similarity-quantifying" measure (Sucholutsky et al., 2023) of representations based on the likelihood of the models.

Let $\boldsymbol{\theta}^*$ and $\boldsymbol{\theta}'$ be two models, and let $\boldsymbol{\theta}^*$ satisfy the diversity condition (Definition 1) for both $\mathbf{f}^*$ and $\mathbf{g}^*$. Let $\mathbf{y}_0, ..., \mathbf{y}_M$ be the $\mathbf{y}_i$s from the diversity condition on $\mathbf{g}^*$. Let $\mathbf{L}^*$ be the matrix with columns $\mathbf{g}^*(\mathbf{y}_i) - \mathbf{g}^*(\mathbf{y}_0)$ and $\mathbf{L}'$ the matrix with columns $\mathbf{g}'(\mathbf{y}_i) - \mathbf{g}'(\mathbf{y}_0)$. Let $\mathbf{x}_0, ..., \mathbf{x}_M$ be the $\mathbf{x}_i$s from the diversity condition on $\mathbf{f}^*$. Let $\mathbf{N}^*$ be the matrix with columns $\mathbf{f}^*(\mathbf{x}_i) - \mathbf{f}^*(\mathbf{x}_0)$ and $\mathbf{N}'$ the matrix with columns $\mathbf{f}'(\mathbf{x}_i) - \mathbf{f}'(\mathbf{x}_0)$. Then

$$\mathbf{f}^*(\mathbf{x}) = \mathbf{A}\mathbf{f}'(\mathbf{x}) + \mathbf{h}_{\mathbf{f}^*}(\mathbf{x}) \tag{38}$$

$$\mathbf{g}^*(\mathbf{y}) = \mathbf{B}\mathbf{g}'(\mathbf{y}) + \mathbf{h}_{\mathbf{g}^*}(\mathbf{y}) \tag{39}$$

where $\mathbf{A} = \mathbf{L}^{*-\top}\mathbf{L}'^\top$, $\mathbf{h}_{\mathbf{f}^*}(\mathbf{x}) = \mathbf{L}^{*-\top}\boldsymbol{\epsilon}_{\mathbf{y}}(\mathbf{x})$, and $\boldsymbol{\epsilon}_{\mathbf{y}}(\mathbf{x})$ is a vector function with each entry equal to $\epsilon_{\mathbf{y}i}(\mathbf{x}) = \mathbf{f}^*(\mathbf{x})^\top\mathbf{g}^*(\mathbf{y}_i) - \mathbf{f}'(\mathbf{x})^\top\mathbf{g}'(\mathbf{y}_i) + \mathbf{f}'(\mathbf{x})^\top\mathbf{g}'(\mathbf{y}_0) - \mathbf{f}^*(\mathbf{x})^\top\mathbf{g}^*(\mathbf{y}_0)$. Also, $\mathbf{B} = \mathbf{N}^{*-\top}\mathbf{N}'^\top$, $\mathbf{h}_{\mathbf{g}}(\mathbf{y}) = \mathbf{N}^{*-\top}(\boldsymbol{\epsilon}_x(\mathbf{y}) + \mathbf{c})$, and $\boldsymbol{\epsilon}_{\mathbf{x}}(\mathbf{y})$ is a vector function with each entry equal to $\epsilon_{xi}(\mathbf{y}) = \mathbf{f}^*(\mathbf{x}_i)^\top\mathbf{g}^*(\mathbf{y}) - \mathbf{f}'(\mathbf{x}_i)^\top\mathbf{g}'(\mathbf{y}) + \mathbf{f}'(\mathbf{x}_0)^\top\mathbf{g}'(\mathbf{y}) - \mathbf{f}^*(\mathbf{x}_0)^\top\mathbf{g}^*(\mathbf{y})$, and $\mathbf{c}$ is a constant vector.

*Proof.* We first show that $\mathbf{f}^*(\mathbf{x}) = \mathbf{A}\mathbf{f}'(\mathbf{x}) + \mathbf{h}_{\mathbf{f}^*}(\mathbf{x})$.

Let the codomains of $\mathbf{f}^*, \mathbf{g}^*, \mathbf{f}', \mathbf{g}'$ be in $\mathbb{R}^M$. Let $\mathbf{y}_0, ..., \mathbf{y}_M$ be the ones which exist according to the diversity condition on $\mathbf{g}^*$. Let $Z^*(\mathbf{x}, \mathcal{S}) = \sum_{\mathbf{y}_j \in \mathcal{S}} \exp(\mathbf{f}^*(\mathbf{x})^\top\mathbf{g}^*(\mathbf{y}_j))$, and similarly for $Z'(\mathbf{x}, \mathcal{S})$. Then

$$p^*(\mathbf{y}|\mathbf{x}, \mathcal{S}) = p^*(\mathbf{y}|\mathbf{x}, \mathcal{S})\frac{p'(\mathbf{y}|\mathbf{x}, \mathcal{S})}{p'(\mathbf{y}|\mathbf{x}, \mathcal{S})}$$

$$\mathbf{f}^*(\mathbf{x})^\top\mathbf{g}^*(\mathbf{y}) - \log(Z^*(\mathbf{x}, \mathcal{S})) = \mathbf{f}^*(\mathbf{x})^\top\mathbf{g}^*(\mathbf{y}) - \log(Z^*(\mathbf{x}, \mathcal{S}))$$
$$+ \mathbf{f}'(\mathbf{x})^\top\mathbf{g}'(\mathbf{y}) - \log(Z'(\mathbf{x}, \mathcal{S}))$$
$$- \mathbf{f}'(\mathbf{x})^\top\mathbf{g}'(\mathbf{y}) + \log(Z'(\mathbf{x}, \mathcal{S}))$$

for all $\mathbf{y}$. In particular, it is true for $\mathbf{y}_0, ..., \mathbf{y}_M$. So we can write up $M + 1$ equations of this kind. If we subtract the equation with $\mathbf{y}_0$ from these, we are left with $M$ equations of the form

$$\mathbf{f}^*(\mathbf{x})^\top \mathbf{g}^*(\mathbf{y}_i) - \mathbf{f}^*(\mathbf{x})^\top \mathbf{g}^*(\mathbf{y}_0) + \log(Z^*(\mathbf{x}, \mathcal{S})) - \log(Z^*(\mathbf{x}, \mathcal{S}))$$

$$= \mathbf{f}^*(\mathbf{x})^\top \mathbf{g}^*(\mathbf{y}_i) - \mathbf{f}^*(\mathbf{x})^\top \mathbf{g}^*(\mathbf{y}_0) + \log(Z^*(\mathbf{x}, \mathcal{S})) - \log(Z^*(\mathbf{x}, \mathcal{S}))$$

$$+ \mathbf{f}'(\mathbf{x})^\top \mathbf{g}'(\mathbf{y}_i) - \mathbf{f}'(\mathbf{x})^\top \mathbf{g}'(\mathbf{y}_0) + \log(Z'(\mathbf{x}, \mathcal{S})) - \log(Z'(\mathbf{x}, \mathcal{S}))$$

$$- (\mathbf{f}'(\mathbf{x})^\top \mathbf{g}'(\mathbf{y}_i) - \mathbf{f}'(\mathbf{x})^\top \mathbf{g}'(\mathbf{y}_0) + \log(Z'(\mathbf{x}, \mathcal{S})) - \log(Z'(\mathbf{x}, \mathcal{S})))$$

$$\mathbf{f}^*(\mathbf{x})^\top \mathbf{g}^*(\mathbf{y}_i) - \mathbf{f}^*(\mathbf{x})^\top \mathbf{g}^*(\mathbf{y}_0) = \mathbf{f}'(\mathbf{x})^\top \mathbf{g}'(\mathbf{y}_i) - \mathbf{f}'(\mathbf{x})^\top \mathbf{g}'(\mathbf{y}_0)$$

$$+ \mathbf{f}^*(\mathbf{x})^\top \mathbf{g}^*(\mathbf{y}_i) - \mathbf{f}'(\mathbf{x})^\top \mathbf{g}'(\mathbf{y}_i)$$

$$+ \mathbf{f}'(\mathbf{x})^\top \mathbf{g}'(\mathbf{y}_0) - \mathbf{f}^*(\mathbf{x})^\top \mathbf{g}^*(\mathbf{y}_0)$$

$$\mathbf{f}^*(\mathbf{x})^\top (\mathbf{g}^*(\mathbf{y}_i) - \mathbf{g}^*(\mathbf{y}_0)) = \mathbf{f}'(\mathbf{x})^\top (\mathbf{g}'(\mathbf{y}_i) - \mathbf{g}'(\mathbf{y}_0))$$

$$+ \mathbf{f}^*(\mathbf{x})^\top \mathbf{g}^*(\mathbf{y}_i) - \mathbf{f}'(\mathbf{x})^\top \mathbf{g}'(\mathbf{y}_i)$$

$$+ \mathbf{f}'(\mathbf{x})^\top \mathbf{g}'(\mathbf{y}_0) - \mathbf{f}^*(\mathbf{x})^\top \mathbf{g}^*(\mathbf{y}_0)$$

$$(\mathbf{g}^*(\mathbf{y}_i) - \mathbf{g}^*(\mathbf{y}_0))^\top \mathbf{f}^*(\mathbf{x}) = (\mathbf{g}'(\mathbf{y}_i) - \mathbf{g}'(\mathbf{y}_0))^\top \mathbf{f}'(\mathbf{x})$$

$$+ \mathbf{f}^*(\mathbf{x})^\top \mathbf{g}^*(\mathbf{y}_i) - \mathbf{f}'(\mathbf{x})^\top \mathbf{g}'(\mathbf{y}_i)$$

$$+ \mathbf{f}'(\mathbf{x})^\top \mathbf{g}'(\mathbf{y}_0) - \mathbf{f}^*(\mathbf{x})^\top \mathbf{g}^*(\mathbf{y}_0)$$

$$(\mathbf{g}^*(\mathbf{y}_i) - \mathbf{g}^*(\mathbf{y}_0))^\top \mathbf{f}^*(\mathbf{x}) = (\mathbf{g}'(\mathbf{y}_i) - \mathbf{g}'(\mathbf{y}_0))^\top \mathbf{f}'(\mathbf{x}) + \epsilon_{yi}(\mathbf{x})$$

where $\epsilon_{yi}(\mathbf{x}) = \mathbf{f}^*(\mathbf{x})^\top \mathbf{g}^*(\mathbf{y}_i) - \mathbf{f}'(\mathbf{x})^\top \mathbf{g}'(\mathbf{y}_i) + \mathbf{f}'(\mathbf{x})^\top \mathbf{g}'(\mathbf{y}_0) - \mathbf{f}^*(\mathbf{x})^\top \mathbf{g}^*(\mathbf{y}_0)$. Note that

$$\epsilon_{yi}(\mathbf{x}) = \log(p^*(\mathbf{y}_i|\mathbf{x}, \mathcal{S})) - \log(p'(\mathbf{y}_i|\mathbf{x}, \mathcal{S})) + \log(p'(\mathbf{y}_0|\mathbf{x}, \mathcal{S})) - \log(p^*(\mathbf{y}_0|\mathbf{x}, \mathcal{S})) \quad (40)$$

so it is a difference of log-likelihoods.

Let $\mathbf{L}^*$ be the matrix which has $\mathbf{g}^*(\mathbf{y}_i) - \mathbf{g}^*(\mathbf{y}_0)$ as columns, let $\mathbf{L}'$ be the matrix which has $\mathbf{g}'(\mathbf{y}_i) - \mathbf{g}'(\mathbf{y}_0)$ as columns and let $\boldsymbol{\epsilon}_y(\mathbf{x})$ be the vector which has $\epsilon_{yi}(\mathbf{x})$ as entries. We can then stack the equations to get

$$\mathbf{L}^{*T} \mathbf{f}^*(\mathbf{x}) = \mathbf{L}'^\top \mathbf{f}'(\mathbf{x}) + \boldsymbol{\epsilon}_y(\mathbf{x}) \quad (41)$$

and since $\mathbf{L}^*$ is invertible,

$$\mathbf{f}^*(\mathbf{x}) = \mathbf{L}^{*-\top} \left( \mathbf{L}'^\top \mathbf{f}'(\mathbf{x}) + \boldsymbol{\epsilon}_y(\mathbf{x}) \right) \quad (42)$$

If we set $A = \mathbf{L}^{*-\top} \mathbf{L}'^\top$ and $\mathbf{h}_{\mathbf{f}^*}(\mathbf{x}) = \mathbf{L}^{*-\top} \boldsymbol{\epsilon}_y(\mathbf{x})$, we get the result. $\qquad \square$

**Notes on when A is invertible**   We see that if the diversity condition is satisfied for $\mathbf{g}'$ with the same $\mathbf{y}_i$ as for $\mathbf{g}^*$, $\mathbf{A}$ is invertible.

If we do not assume that $\mathbf{g}'$ satisfies the diversity condition, we can use the diversity condition on $\mathbf{f}^*$ to pick points $\mathbf{x}_0, ..., \mathbf{x}_M$ such that $\mathbf{f}^*(\mathbf{x}_i) - \mathbf{f}^*(\mathbf{x}_0)$ are linearly independent. Let $\mathbf{N}^*$ be the matrix with $\mathbf{f}^*(\mathbf{x}_i) - \mathbf{f}^*(\mathbf{x}_0)$ as columns, let $\mathbf{N}'$ be the matrix with $\mathbf{f}'(\mathbf{x}_i) - \mathbf{f}'(\mathbf{x}_0)$ as columns and let $\mathbf{E}$ be the matrix with $\boldsymbol{\epsilon}_y(\mathbf{x}_i) - \boldsymbol{\epsilon}_y(\mathbf{x}_0)$ as columns. Then we have

$$\mathbf{N}^* = \mathbf{L}^{*-\top} \left( \mathbf{L}'^\top \mathbf{N}' + \mathbf{E} \right) \quad (43)$$

If $\mathbf{E} = \mathbf{0}$, we get $\mathbf{N}^* = \mathbf{A}\mathbf{N}'$. Since we know that for any two matrices, $\mathbf{B}, \mathbf{C}$, $\text{rank}(\mathbf{BC}) \leq \min(\text{rank}(\mathbf{B}), \text{rank}(\mathbf{C}))$, and $\mathbf{N}^*$ has rank $M$, we see that $\mathbf{A}$ and $\mathbf{N}'$ must both also have rank $M$. Thus, $\mathbf{A}$ and $\mathbf{N}'$ are invertible.

If $\mathbf{E} \neq \mathbf{0}$, we get from the same argument that $\mathbf{L}'^\top \mathbf{N}' + \mathbf{E}$ is an invertible matrix. However, this does not give us invertibility of $\mathbf{L}'$ or $\mathbf{N}'$.

*Proof.* Next we show that $\mathbf{g}^*(\mathbf{y}) = \mathbf{B}\mathbf{g}'(\mathbf{y}) + \mathbf{h}_\mathbf{g}(\mathbf{y})$.

As before we have that

$$\mathbf{f}^*(\mathbf{x})^\top \mathbf{g}^*(\mathbf{y}) - \log(Z^*(\mathbf{x}, \mathcal{S})) = \mathbf{f}'(\mathbf{x})^\top \mathbf{g}'(\mathbf{y}) - \log(Z'(\mathbf{x}, \mathcal{S})) \tag{44}$$

$$+ \mathbf{f}^*(\mathbf{x})^\top \mathbf{g}^*(\mathbf{y}) - \log(Z^*(\mathbf{x}, \mathcal{S})) \tag{45}$$

$$- \mathbf{f}'(\mathbf{x})^\top \mathbf{g}'(\mathbf{y}) + \log(Z'(\mathbf{x}, \mathcal{S})) \tag{46}$$

holds for all $\mathbf{x}$. In particular, it is true for the $M + 1$ $\mathbf{x}$s, $\mathbf{x}_0, ..., \mathbf{x}_M$ which exist according to the diversity condition on $\mathbf{f}^*$. So we can write up $M + 1$ equations of this kind. If we subtract the equation with $\mathbf{x}_0$ from these, we are left with $M$ equations of the form

$$\mathbf{f}^*(\mathbf{x}_i)^\top \mathbf{g}^*(\mathbf{y}) - \mathbf{f}^*(\mathbf{x}_0)^\top \mathbf{g}^*(\mathbf{y}) + \log(Z^*(\mathbf{x}_0, \mathcal{S})) - \log(Z^*(\mathbf{x}_i, \mathcal{S}))$$

$$= \mathbf{f}'(\mathbf{x}_i)^\top \mathbf{g}'(\mathbf{y}) - \mathbf{f}'(\mathbf{x}_0)^\top \mathbf{g}'(\mathbf{y}) + \log(Z'(\mathbf{x}_0, \mathcal{S})) - \log(Z'(\mathbf{x}_i, \mathcal{S}))$$

$$+ \mathbf{f}^*(\mathbf{x}_i)^\top \mathbf{g}^*(\mathbf{y}) - \mathbf{f}^*(\mathbf{x}_0)^\top \mathbf{g}^*(\mathbf{y}) + \log(Z^*(\mathbf{x}_0, \mathcal{S})) - \log(Z^*(\mathbf{x}_i, \mathcal{S}))$$

$$- \mathbf{f}'(\mathbf{x}_i)^\top \mathbf{g}'(\mathbf{y}) + \mathbf{f}'(\mathbf{x}_0)^\top \mathbf{g}'(\mathbf{y}) - \log(Z'(\mathbf{x}_0, \mathcal{S})) + \log(Z'(\mathbf{x}_i, \mathcal{S}))$$

$$(\mathbf{f}^*(\mathbf{x}_i) - \mathbf{f}^*(\mathbf{x}_0))^\top \mathbf{g}^*(\mathbf{y}) = (\mathbf{f}'(\mathbf{x}_i) - \mathbf{f}'(\mathbf{x}_0))^\top \mathbf{g}'(\mathbf{y}) + \epsilon_{xi}(\mathbf{y}) + c_i$$

where

$$\epsilon_{xi}(\mathbf{y}) = \mathbf{f}^*(\mathbf{x}_i)^\top \mathbf{g}^*(\mathbf{y}) - \mathbf{f}'(\mathbf{x}_i)^\top \mathbf{g}'(\mathbf{y}) + \mathbf{f}'(\mathbf{x}_0)^\top \mathbf{g}'(\mathbf{y}) - \mathbf{f}^*(\mathbf{x}_0)^\top \mathbf{g}^*(\mathbf{y}) \tag{47}$$

and

$$c_i = \log(Z^*(\mathbf{x}_0, \mathcal{S})) - \log(Z^*(\mathbf{x}_i, \mathcal{S})) - \log(Z'(\mathbf{x}_0, \mathcal{S})) + \log(Z'(\mathbf{x}_i, \mathcal{S})) \tag{48}$$

$$+ \log(Z'(\mathbf{x}_0, \mathcal{S})) - \log(Z'(\mathbf{x}_i, \mathcal{S})) \tag{49}$$

$$= \log\left(\frac{Z^*(\mathbf{x}_0, \mathcal{S})}{Z'(\mathbf{x}_0, \mathcal{S})}\right) + \log\left(\frac{Z'(\mathbf{x}_i, \mathcal{S})}{Z^*(\mathbf{x}_i, \mathcal{S})}\right) + \log\left(\frac{Z'(\mathbf{x}_0, \mathcal{S})}{Z'(\mathbf{x}_i, \mathcal{S})}\right) \tag{50}$$

which is a constant.

Let $\mathbf{N}^*$ be the matrix with $\mathbf{f}^*(\mathbf{x}_i) - \mathbf{f}^*(\mathbf{x}_0)$ as columns, let $\mathbf{N}'$ be the matrix with $\mathbf{f}'(\mathbf{x}_i) - \mathbf{f}'(\mathbf{x}_0)$ as columns, let $\mathbf{c}$ be the vector with $c_i$ as entries and let $\boldsymbol{\epsilon}_x(\mathbf{y})$ be the vector which has $\epsilon_{xi}(\mathbf{y})$ as entries. Then

$$\mathbf{N}^{*T} \mathbf{g}^*(\mathbf{y}) = \mathbf{N}'^\top \mathbf{g}'(\mathbf{y}) + \epsilon_{xi}(\mathbf{y}) + \mathbf{c} \tag{51}$$

$$\mathbf{g}^*(\mathbf{y}) = \mathbf{N}^{*-\top}(\mathbf{N}'^\top \mathbf{g}'(\mathbf{y}) + \mathbf{c} + \boldsymbol{\epsilon}_x(\mathbf{y})) \tag{52}$$

Letting $\mathbf{B} = \mathbf{N}^{*-\top} \mathbf{N}'^\top$ and $\mathbf{h_g}(\mathbf{y}) = \mathbf{N}^{*-\top}(\boldsymbol{\epsilon}_x(\mathbf{y}) + \mathbf{c})$, we have the result.

$\square$

# D   Full Example of Close to Zero Loss where Representations are Dissimilar

In this example, we have $M = 2$ and a classification task with four labels, $\mathbf{y}_0, \mathbf{y}_1, \mathbf{y}_2, \mathbf{y}_3$. The example relies on the fact that if we fix non-zero angles between the unembedding vectors, and we let the embedding representations be closer to the correct label in terms of angle than the incorrect ones, then we can make the likelihood of the correct label arbitrarily close to 1 by only changing the lengths of the unembedding vectors. We will first describe the unembedding vectors, $\mathbf{g}(\mathbf{y}_i)$, and then describe the embedding representations, $\mathbf{f}(\mathbf{x})$, based on the $\mathbf{g}(\mathbf{y}_i)$s. For model $\boldsymbol{\theta}'$, we let $\|\mathbf{g}'(\mathbf{y}_i)\| = 1100$ for all $i$ (see Appendix E for how this was chosen) and the angle between $\mathbf{g}'(\mathbf{y}_i)$ and $\mathbf{g}'(\mathbf{y}_{i+1})$ be $\pi/16$ radians, starting with $\mathbf{g}'(\mathbf{y}_0) = (1100, 0)$. So we have $\mathbf{g}'(\mathbf{y}_1) = (1100 \cdot \cos(\pi/16), 1100 \cdot \sin(\pi/16))$, $\mathbf{g}'(\mathbf{y}_2) = (1100 \cdot \cos(2\pi/16), 1100 \cdot \sin(2\pi/16))$, $\mathbf{g}'(\mathbf{y}_3) = (1100 \cdot \cos(3\pi/16), 1100 \cdot \sin(3\pi/16))$.

To generate our $\mathbf{f}'(\mathbf{x})$ representations, we draw 1000 samples for each label from a uniform distribution from $[-\pi/128, \pi/128]$. This represents the angle between $\mathbf{f}'(\mathbf{x})$ and the representation of the correct label $\mathbf{g}'(\mathbf{y}_i)$. To get the length of the $\mathbf{f}'(\mathbf{x})$, we draw 4000 samples from a standard normal distribution, $z \in \mathcal{N}(0, 1)$, and transform it to be a value larger than 1 in the following way: $\|\mathbf{f}'(\mathbf{x})\| = |z| + 1$. See a visualization of the $\mathbf{f}'(\mathbf{x})$ for model 1 in Fig. 2.

For model $\boldsymbol{\theta}^*$, we let $\|\mathbf{g}^*(\mathbf{y}_i)\| = 100$ for all $i$, the angle between $\mathbf{g}^*(\mathbf{y}_0)$ and $\mathbf{g}^*(\mathbf{y}_1)$ will be $2\pi/6$ radians and the angle between $\mathbf{g}^*(\mathbf{y}_2), \mathbf{g}^*(\mathbf{y}_3)$ and $\mathbf{g}^*(\mathbf{y}_3), \mathbf{g}^*(\mathbf{y}_0)$ be $\pi/2$ radians, starting with

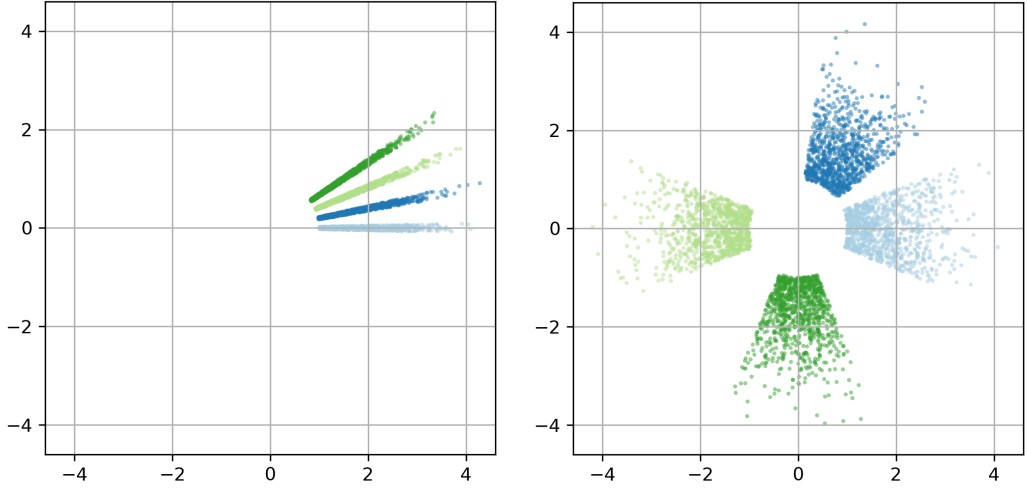

Figure 2: $\mathbf{f}'(\mathbf{x})$ for model $\boldsymbol{\theta}'$         Figure 3: $\mathbf{f}^*(\mathbf{x})$ for model $\boldsymbol{\theta}^*$

$\mathbf{g}^*(\mathbf{y}_0) = (100, 0)$. So we have $\mathbf{g}^*(\mathbf{y}_1) = (100 \cdot \cos(2\pi/6), 100 \cdot \sin(2\pi/6))$, $\mathbf{g}^*(\mathbf{y}_2) = (-100, 0)$, $\mathbf{g}^*(\mathbf{y}_3) = (0, -100)$.

To generate our $\mathbf{f}^*(\mathbf{x})$ representations, we take the samples we drew from the uniform distribution for the first model and multiply them with 16, so we get samples from a uniform distribution from $[-\pi/8, \pi/8]$. These represent the angle between $\mathbf{f}^*(\mathbf{x})$ and the representation of the correct label $\mathbf{g}^*(\mathbf{y}_i)$. We use the same lengths for the $\mathbf{f}^*(\mathbf{x})$s as in model 1. See visualization in Fig. 3.

We can now calculate the negative log-likelihood ($NLL$) for these two models using Eq. (1). For model $\boldsymbol{\theta}'$, we get $NLL' \approx 9 \cdot 10^{-10}$ and for model $\boldsymbol{\theta}^*$, we get $NLL^* \approx 7 \cdot 10^{-10}$. In fact, for the correct labels, we get a small difference in log-likelihood for all datapoints. The maximal difference for the two models is $8 \cdot 10^{-7}$. As mentioned above, we could make this loss arbitrarily small, by increasing the lengths of the unembeddings.

Now both of these models have close to zero loss, and their loss thus is close to equal: however, when we calculate the mean canonical correlation between the embedding representations, we get $m_{\text{CCA}}(\mathbf{f}'(\mathbf{x}), \mathbf{f}^*(\mathbf{x})) \approx 0.42$, which is very far from the value of 1 which would indicate a perfect linear relationship. To get an idea of how dissimilar this is, let $\mathbf{f}'_{noise}(\mathbf{x})$ be $\mathbf{f}'(\mathbf{x})$, where we have added Gaussian noise to each dimension with four times the variance that $\mathbf{f}'(\mathbf{x})$ has in that dimension. We then get $m_{\text{CCA}}(\mathbf{f}'(\mathbf{x}), \mathbf{f}'_{noise}(\mathbf{x})) \approx 0.44$. If we only add noise with twice the variance, we get $m_{\text{CCA}}(\mathbf{f}'(\mathbf{x}), \mathbf{f}'_{noise}(\mathbf{x})) \approx 0.57$. Thus, we can see that the representations of the models are quite far from being similar.

We can also calculate $\mathbf{L}'^{\top}\mathbf{f}'(\mathbf{x})$ and $\epsilon_y(\mathbf{x})$ (the equivalent of Eq. (6)) for the models and consider the relative size of $\epsilon_y(\mathbf{x})$ compared to $\mathbf{L}'^{\top}\mathbf{f}'(\mathbf{x})$ in each dimension. We use $\mathbf{y}_0, \mathbf{y}_1$ and $\mathbf{y}_3$ for the diversity condition and let

$$\mathbf{L}'^{\top} = \begin{bmatrix} (\mathbf{g}'(\mathbf{y}_1) - \mathbf{g}'(\mathbf{y}_0))^{\top} \\ (\mathbf{g}'(\mathbf{y}_3) - \mathbf{g}'(\mathbf{y}_0))^{\top} \end{bmatrix} \tag{53}$$

Let $\mathbf{L}'^{\top}\mathbf{f}'(\mathbf{x})_i$ be the $i$'th dimension of $\mathbf{L}'^{\top}\mathbf{f}'(\mathbf{x})$. Then

$$\mathbb{E}_{\mathbf{x} \in \mathcal{X}} \left[ \frac{|\epsilon_{y1}(\mathbf{x})|}{|\mathbf{L}'^{\top}\mathbf{f}'(\mathbf{x})_1|} \right] = 2.2 \quad \text{and} \quad \mathbb{E}_{\mathbf{x} \in \mathcal{X}} \left[ \frac{|\epsilon_{y2}(\mathbf{x})|}{|\mathbf{L}'^{\top}\mathbf{f}'(\mathbf{x})_2|} \right] = 0.8$$

This means that on average the value of the "error term" in one of the dimensions is more than twice as large as the representation term and thus it makes sense that the representations are not close to being linear transformations of each other.

# E  Choosing the Lengths of the Embedding and Unembedding Vectors

To decide on the lengths of the embedding and unembedding vectors, in the example, we first noted that

$$\mathbf{f}(\mathbf{x})^\top \mathbf{g}(\mathbf{y}) = \cos(\mathbf{f}(\mathbf{x}), \mathbf{g}(\mathbf{y})) \|\mathbf{f}(\mathbf{x})\| \|\mathbf{g}(\mathbf{y})\| \tag{54}$$

Therefore, if we decide on the angles for the $\mathbf{g}(\mathbf{y})$s and $\mathbf{f}(\mathbf{x})$s and let all the $\mathbf{g}(\mathbf{y})$s have the same length, we can write the log-likelihood as a function of $v = \|\mathbf{f}(\mathbf{x})\| \|\mathbf{g}(\mathbf{y})\|$

$$\log p_\theta(\mathbf{y}|\mathbf{x}, \mathcal{S}) = \mathbf{f}_\theta(\mathbf{x})^\top \mathbf{g}_\theta(\mathbf{y}) - \log \left( \sum_{\mathbf{y}' \in \mathcal{S}} \exp(\mathbf{f}_\theta(\mathbf{x})^\top \mathbf{g}_\theta(\mathbf{y}')) \right) \tag{55}$$

$$= \cos(\mathbf{f}(\mathbf{x}), \mathbf{g}(\mathbf{y})) \cdot v - \log \left( \sum_{\mathbf{y}' \in \mathcal{S}} \exp(\cos(\mathbf{f}(\mathbf{x}), \mathbf{g}(\mathbf{y})) \cdot v) \right) \tag{56}$$

Inspecting this function (see Fig. 4), we see we can get arbitrarily close to zero log-likelihood, by increasing $v$. Thus, for models where the $\mathbf{g}(\mathbf{y})$s are closer in terms of angles, we can simply increase the length of the $\mathbf{g}(\mathbf{y})$s.

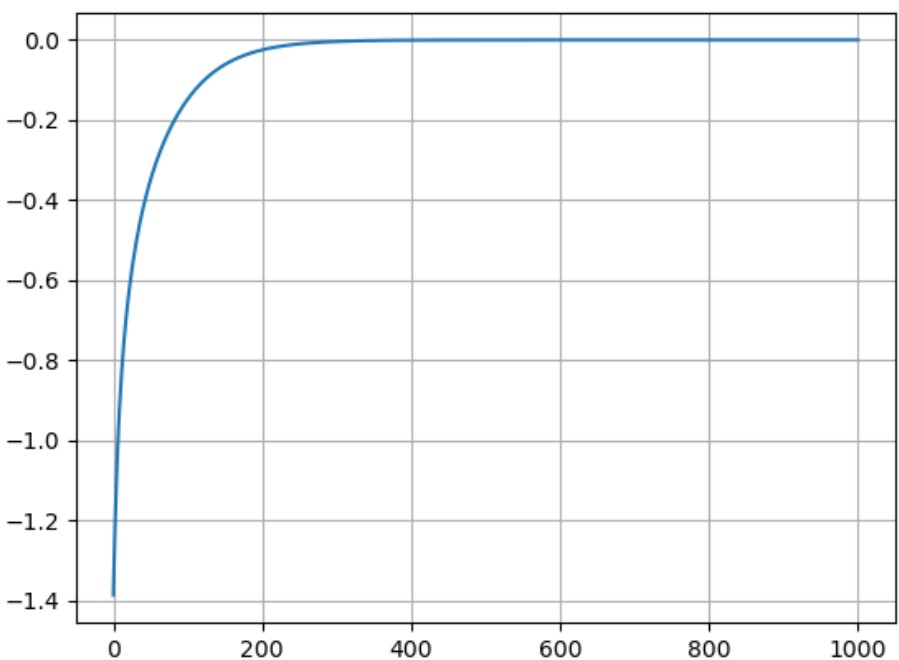

Figure 4: log-likelihood as function of product of vector lengths for fixed angles

