# OpenReview forum: "Challenges in Explaining Representational Similarity through Identifiability"
_NeurIPS.cc/2024/Workshop/UniReps — UniReps_

### Official Review · Reviewer_H7Yu · 2024-10-01
**Review of Challenges in Explaining Representational Similarity through Identifiability**

**Rating:** 8
**Confidence:** 4

**Review:**

- I think this paper makes a nice point: two models with similar losses needn't have similar likelihoods for all pairs of inputs and outputs, and thus (at least in this particular model class) this means that two models with nearly identical performance needn't have representations that are linear transformations of one another
- I like the constructed example, but I wonder if some other example is possible with an even lower mean CCA score?
- What's missing to me is how meaningful this observation is in practice on real data
- nit: Please use `\citep{}` and `\citet{}` as appropriate in the context

---

### Official Review · Reviewer_eN3W · 2024-10-06
**The paper addresses challenges in explaining representational similarity through identifiability theory but lacks empirical validation and practical solutions.**

**Rating:** 5
**Confidence:** 4

**Review:**

The paper addresses the question of "Why different models often produce similar data representations?". Focusing on the challenges of explaining this phenomenon through identifiability theory, the authors identify two key obstacles: First, they highlight that the assumption of exact likelihood equality is rarely met in practice. To address this, they extend existing identifiability results by relating deviations in representations to differences in log-likelihoods. Second, they demonstrate that even models with near-optimal and similar loss values can produce highly dissimilar representations due to the distinction between loss and likelihood.

Although the paper successfully identifies key challenges, it lacks concrete solutions or empirical evaluations to test its theoretical claims. Including experiments or practical suggestions on how to overcome the highlighted obstacles would have provided a more well-rounded contribution. Additionally, the paper could benefit from more detailed discussions on the potential implications of the findings for broader representation learning scenarios, particularly for models trained on different datasets or with varying objectives.

---

### Official Review · Reviewer_D6eA · 2024-10-06
**Interesting and substantial work**

**Rating:** 8
**Confidence:** 3

**Review:**

**Pros:**
* This work is high quality and novel.
* This paper provides significant and interesting theoretical results for representational similarity, and reveal open questions/problems for future research.
* The toy example provided in Section 4 showing that two models with close-to-equal losses produce very different representations is interesting and helpful.

**Cons:**
* While most of the paper is clear, there is a critical component that was not clearly defined. Under section 2, $g^*$ and $g'$ are introduced as unembedding functions. But what are unembedding functions? Do they map the labels $y$ to embeddings $z$? Given the importance of unembedding functions in the key results of the paper, I think it would be helpful to clearly explain what they are.

---

### Official Review · Reviewer_BakA · 2024-10-07
**The authors have highlighted the shortcomings of applying Identifiability theorem to two different models with near-optimal loss value. The work is of great interst as it could be employed in enhancing the understanding of model's representations.**

**Rating:** 7
**Confidence:** 2

**Review:**

### Summary
The paper highlights the shortcomings in understanding representational similarity between two models using the Identifiability theory:
- The satisfiability of the key assumption that the two models should have the same log-likelihood is rare.
- The two converged models with near-optimal loss values may not have similar representations.

### Strengths
The mentioned key challenges are corroborated by -
- A theoretical proof of the theorem that representation obtained by one model can be written as a sum of a linear transformation of the representations obtained by another model and an error term that vanishes if two models have the same likelihood for all x and y.
- An experimental example to show that two models with near-optimal loss value can result in dissimilar representations.

### Weakness
The experiment is shown on representations that are manually created to achieve near-optimal loss. However, it would be better to verify this for neural networks trained on similar datasets with near-optimal loss value but with different initialization.

### Significance, clarity, novely
The paper highlights the shortcomings of the Identifiability theory. The work is of great significance as it could be leveraged to enhance the model's interpretability. The paper is well-written with sufficient originality.

---

### Author Response · Authors · 2024-10-16
**Thank you to the Reviewers and Edits**

We would like to thank all the reviewers for their comments.

In response to the reviews, we have

- Added some explanation of the embedding and unembeddings functions in section 2 under "Model Class".
- Added a the end of Section 4 how it is possible to construct an example with even lower m_CCA
- Changed the use of citet and citep

We agree with the reviewers that is is a good idea to run experiments to show how something like the constructed example can occur in practice, and we will do this in future work.

---

### Decision · Program_Chairs · 2024-10-10

**Decision:**

Accept

**Comment:**

In light of the positive reviewers' feedback and relevancy of the submission, we are pleased to accept this paper for presentation at UniReps 2024. We kindly ask the authors to incorporate the reviewers' suggestions and feedback in the final camera-ready version of the manuscript.